# An Exploratory Study on the Effects of Souchard Postural Gymnastics in Parkinson’s Disease Patients with Camptocormia: A Quasi-Experimental Approach

**DOI:** 10.3390/jcm13206166

**Published:** 2024-10-16

**Authors:** Emanuele Amadio, Matteo Mencio, Alessandra Carlizza, Francescaroberta Panuccio, Giovanni Sellitto, Ilaria Ruotolo, Rachele Simeon, Anna Berardi, Giovanni Galeoto

**Affiliations:** 1Department of Human Neurosciences, Sapienza University of Rome, Viale dell’Università, 30, 00185 Rome, Italy; emanueleama00@gmail.com (E.A.); francescaroberta.panuccio@uniroma1.it (F.P.); giovanni.sellitto@uniroma1.it (G.S.); ilaria.ruotolo@uniroma1.it (I.R.); rachele.simeon@uniroma1.it (R.S.); anna.berardi@uniroma1.it (A.B.); 2UniCamillus University of Rome, Via di Sant’Alessandro, 8, 00131 Rome, Italy; matteo.25.98@gmail.com (M.M.); alessandra.carlizza@unicamillus.org (A.C.); 3IRCCS Neuromed, Via Atinense, 18, 86077 Pozzilli, Italy

**Keywords:** rehabilitation, Parkinson’s disease, camptocormia, postural gymnastic, treatment

## Abstract

**Background/Objective**: Parkinson’s disease (PD), a prevalent neurodegenerative disorder, leads to motor and non-motor impairments, affecting quality of life. Camptocormia can be one of the motor signs of PD, characterized by a severe and abnormal forward flexion of the thoracolumbar spine that typically occurs when walking or standing. The following study aims to verify whether postural gymnastics can be an effective treatment for trunk control, balance, activities of daily living, and general well-being in patients with early-stage PD and camptocormia. **Methods**: Nine participants (mean age 67.7 ± 7.8) with early PD (Hoehn and Yahr Scale ≤ 2) received 10 biweekly physiotherapy sessions. Outcomes were measured using the Parkinson’s Disease Questionnaire (PDQ-39) and Berg Balance Scale (BBS) along with trunk mobility and muscle tests according to the Medical Research Council (MRC) scale. **Results**: Statistically significant results were noted in the PDQ-39 mobility, ADLs and emotional well-being subscales and in the BBS; statistically significant improvements were also seen in trunk mobility and muscle strength. **Conclusions**: This study shows that the postural gymnastic treatment, according to Souchard, in patients with PD’s camptocormia has obtained good results and has the potential timprove mobility and balance, encouraging and motivating patients in their rehabilitation journeys.

## 1. Introduction

Parkinson’s disease (PD), the second most common neurodegenerative disorder globally, impacts around 7 million individuals across the world, with an annual occurrence of 13.5 cases per 100,000 people [1,2,3].

A majority of those affected are aged 60 or older, and the prevalence of Parkinson’s disease increases with age, with a slightly higher incidence in men compared to women [4]. The main reason that causes this pathology is the loss of dopaminergic neurons in the region of the pars compacta in the substantia nigra, which are fundamental for motor activity in the Central Nervous System (CNS), which will cause a reduction in the thalamus and motor cortex function [4]. According to the International Parkinson and Movement Disorder Society Task Force guidelines, the latest diagnostic criteria focus on motor and non-motor symptoms, including recognition of prodromal phases (e.g., symptoms such as sleep disturbances, olfactory dysfunction, depression, etc.) [5]. Among the primary clinical signs of PD are motor disorders, including bradykinesia, rest tremor, rigidity, and postural instability. The motor symptoms usually start from one side of the body and then affect the contralateral side within a few years [6]. During the progression of the disease, the motor symptoms will cause the patients to develop postural disorders with a specific postural attitude. The main postural disorders are camptocormia, with a kyphotic flexion of at least 45°, and Pisa syndrome, with lateral trunk deviation; both are present during the orthostatic position [7,8,9]. Abnormal posture is a recurrent feature in PD; more than 30% of patients show a neck, trunk, or limb deformity. Camptocormia is an axial postural deformity characterized by an abnormal flexion of the thoracolumbar part of the spine; it induces functional impairment with trunk movement restrictions, balance impairment, gait cycle difficulty, and body image alteration [10]. These changes are evident in the perception of space, proprioception, balance, and ambulation. Strong evidence suggests that non-motor symptoms can be an earlier disease manifestation for PD, appearing years before the first presence of motor symptoms, and they usually include visual problems, cognitive dysfunction, constipation, sleep behavioral disorder (RBD), and mood disturbance [6]. Visual problems due to retinal dopamine depletion, camptocormia, and postural instability increase the risk of falls and fall-related injuries [11]. The primary general treatment commonly employed in the management of Parkinson’s disease includes pharmacological interventions and physical therapy to support the patient, slow down the degenerative process, and maintain their daily functioning in their current conditions. Alternative treatments for Parkinson’s disease encompass virtual reality [12], treadmill training [13], transcranial stimulation of the primary motor cortex [14], and dynamic exercises [15]. In regards to postural interventions and management of secondary symptoms such as chronic pain, gait disorders, and risk of falls, physiotherapy is one of the preferred treatments [16]. However, there are no guidelines, and various interventions are proposed in the scientific literature, such as postural exercises, education, and orthotics.

The rehabilitative technique employed in this study is global postural re-education (GPR) according to the Souchard method [17]. GPR considers the anatomical, metabolic, and functional differences between static and dynamic muscle fibers, and, according to its principles, static muscle function is the leading cause of incorrect postural attitudes and associated mechanical dysfunctions. GPR aims to counteract the pathological muscular mechanism that leads to tendon retraction and hypertonicity by using active postures involving various muscle chains throughout the body and employing eccentric or isometric muscle contractions at increasing lengths, thus achieving balanced control of different muscle groups in the body [18].

The following study hypothesizes that Souchard’s postural gymnastics, which have demonstrated effectiveness in patients with musculoskeletal disorders and certain neurological diseases [18,19,20,21,22], will yield comparable outcomes in terms of improvements in trunk control, balance, and overall quality of life in people with early-stage PD. Although research on GPR for neurological patients is limited, the success observed in the current scientific literature justifies the application of this intervention in the PD population.

The following study aims to verify whether early interventions with postural gymnastics can effectively manage the trunk, balance, activities of daily living (ADLs), and general well-being in patients with early-stage PD and camptocormia.

## 2. Materials and Methods

This study was conducted by the R.E.S. (Ricerca Evidenza e Sviluppo) research group from the Sapienza University of Rome (Italy); in the last few years, the R.E.S. group has been involved in carrying out systematic reviews and validating outcome measures [23,24,25,26,27,28,29,30].

A quasi-experimental study design was chosen because of the expected small sample size and the inability to perform adequate randomization [31,32].

### 2.1. Participants

The inclusion criteria and exclusion criteria of this study are participants, without differentiation of gender and concomitant pathologies, diagnosed with Parkinson’s Disease, stage of disease according to H&Y scale 1 (unilateral involvement with minimal or no functional disability) and 2 (bilateral or midline involvement without impairment of balance). The nine participants were recruited at the Department of Human Neurosciences “Policlinico Umberto I” in Rome between May 2023 and October 2023.

### 2.2. Assessment Tools

The primary outcomes considered in this study are the improvement of camptocormia, postural disorder, trunk mobility, perception of the body in space, and balance. The scales administered in this study are the Berg Balance Scale (BBS) (BBS) [33,34,35], the Parkinson’s Disease Questionnaire-39 (PDQ-39) [36,37,38], and the Medical Research Council (MRC) scale [39]. The individual responsible for conducting the pre- and post-intervention assessments was independent from the individual who administered the rehabilitation treatment.

The Berg Balance Scale (BBS) was used to evaluate the balance of the patients. This scale is based on 14 items of evaluation which are very important for totally understanding the full capacities of the patients in regards to balancing themselves during specific actions. The 14 items score from 0 to 4, where 4 is the best result, for a total of 56. Through the tool’s administration, we can understand if there were any changes at the end of the intervention and after one and three months from the score they had at the beginning. The different results are as follows: if the score exceeds 45, the patient can have a safe gait cycle and balance using aids. If the score is higher than 45, the patient can perform a safe gait cycle and balance without aids. If the total score is under 25, it means that there are severe problems with balance in both dynamic and static conditions, and this is the worst result possible.

The Parkinson’s Disease Questionnaire-39 (PDQ-39) was developed by Peto et al. to evaluate the QoL of the patient. This scale consists of 39 items, with five answers for each question: never, occasionally, sometimes, often, and always. The scale is subdivided into eight subscales: mobility (10 items), ADL (6 items), emotional well-being (6 items), stigma (4 items), social support (3 items), cognitive faculties (4 items), communications (3 items), and bodily discomfort (3 items). Through the administration of the questionnaire, we can understand if there were any changes at the end of the intervention. After one and three months from the score, they had at the beginning where the best result are 0 and the worst is 156, which is the maximum score achievable.

The Medical Research Council (MRC) scale is a 6-item scale for measuring muscle strength through the observation of movements on a scale from 0 (no contraction) to 5 (normal strength). It can be used to monitor muscular function and recovery.

### 2.3. Intervention

The intervention program in this kind of rehabilitative approach is based on 50–60 min of physiotherapy that take place twice a week, with a total of 10 sessions. The main short-term goals of the 10 sessions are as follows: (1) to prevent or improve camptocormia, (2) improve postural disorders, (3) improve trunk mobility, (4) improve proprioception, and (5) improve balance. The long-term goals are to maintain these improvements over time, one and three months after the specific treatment. Before starting physiotherapy, the BBS and PDQ-39 were administered to assess the patient’s current situation regarding balance and to understand how much the patients are active during ADLs. Then, a functional examination of the trunk range of motion and muscle strength through the administration of the Medical Research Council (MRC) scale was conducted.

The examination starts with an informal observation of the patient and an objective clinical examination which is divided into three activities: (1) posture observation on the three planes of space (sagittal, coronal, and transverse plane), (2) trunk-mobility measurements, (3) muscle-strength measurements. The first step of the clinical examination identifies any postural compensations that will be the focus of the rehabilitation treatment. The evaluation of trunk mobility comprises 11 measurements and is conducted using a tape measure after removing patient’s shoes [40]. These are the observed trunk’s ranges of motion (ROMs):Neutral (from C1 to L5) in upright position.Flexion (from C1 to L5) in upright position. The patient is asked to flex forward the body without flexing their knee or using some compensation.Extension (from C1 to L5) in upright position. The patient is asked to extend the back backward.Neutral side bending right in upright position. Measurement is taken from the ground to the hand.Neutral side bending left in upright position. Measurement is taken from the ground to the hand.Right-side bending in upright position. The patient is asked to tilt the trunk on the left side without use any kind of compensation. Measurement is taken from the ground to the hand.Left-side bending in upright position. The patient is asked to tilt the trunk on the left side without use any kind of compensation. Measurement is taken from the ground to the hand.Neutral right rotation, patient in sitting position. Measurement is taken from the upper part of the shoulder to the greater trochanter of the opposite femur.Neutral left rotation, patient in sitting position. Measurement is taken from the upper part of the shoulder to the greater trochanter of the opposite femur.Right rotation, patient in sitting position. Measurement is taken from the upper part of the shoulder to the greater trochanter of the opposite femur;Left rotation, patient in sitting position. Measurement is taken from the shoulder’s upper part to the opposite femur’s greater trochanter.

Finally, the muscle strength of the following muscles was measured according to the MRC scale: (1) rectus abdominis; (2) internal and external obliques; (3) back extensor.

### 2.4. Treatment

The goals of the following study were established in regard to the short-term and long-term prospective. A total of ten 50–60 min sessions of physiotherapy were established and took place twice a week. The proposed treatment was based on postural gymnastics according to the Souchard method. The patient was asked to maintain 2 postures for 30 min:(1)The frog on the ground (see Figure 1). The patient is in supine position with their knees bent and dropped outward and their feet together and flat on the floor; their arms can be placed at their sides or directed by the therapist. Teaching diaphragmatic breathing is essential and crucial for postural gymnastics. A sacrum traction is then performed, allowing the sacrum to create tension along the spine and stretch the extensor muscles. It is crucial to closely observe the patient during the posture to identify any compensations they might be making, allowing us to target and address the muscles that require stretching. Cervical traction stretches the shortened muscles in the anterior neck chain, particularly the sternocleidomastoid and scalene muscles, which are often shortened in PD patients, leading to head flexion [41]. Progressing in the posture involves gradually extending the knees and progressively opening the coxo-femoral angle, allowing for, in the first phase, a lengthening of the pubic adductors, great adductor, and anterior leg muscles before a lengthening the psoas-iliac and the rectus anterior. The posture concludes when the patient’s legs are fully extended on the couch.(2)The frog on the air (see Figure 2). The starting position is the same of the first position; however, the patient’s legs must be gently lifted off the bed, allowing for the maintenance of a coxo-femoral closure that allows us to stretch the posterior muscle chains of the neck, back, and legs. Once the patient is in position, the Souchard rope must be wrapped around the ankles to support the legs, asking the patient to maintain dorsiflexion of the ankles and external rotation of the hip. This enables the reversal of hip rotation, which is essential for transitioning from kinetic to static coordination. The muscle stretching techniques are the same as those used in the first position, remembering to check for any potential detachment of the sacrum from the couch and closely observing the patient during the posture to identify any compensation. Stretching of the trapezius muscle and the pectoralis minor can also be performed to avoid shoulder elevation and anteversion.

### 2.5. Data Analysis

Data were collected and organized through the evaluation of the patients at the beginning (t0) and at the end (t1) of treatment, with one month (t2) and three month (t3) follow ups also taking place. A descriptive analysis of patients personal and general information (gender, age, years of diagnosis, and Hoehn and Yahr (H&Y) score) was then performed. An inferential analysis was then used to determine if there are statistically significant data thanks to the SSPS program, with significance levels being set at a *p*-value less than or equal to 0.05 (*p* ≤ 0.05).

## 3. Results

According to the inclusion criteria, the study’s nine participants gave their informed consent after being informed about the aims and procedures of the following study. The sample was composed of five males and four females with a mean age (±Standard Deviation) of 67.7 ± 7.8 and a stage of H&Y ≤ 2. All patients had concluded the 10 rehabilitative treatment sessions. The results of data are shown in four different tables and represent the data achieved at the beginning (t0) and at the end of the treatment (t1), as well as after one month (t2) and after three months (t3).

The inferential analysis reported the mean, median, standard deviation, and *p*-value. Fundamental is the median of data that is statistically significant and the *p*-value. Results from the administration of the PDQ-39 are reported in Table 1, while Table 2 reports data from the administration of the BBS.

Table 3 and Table 4 report the data, respectively, regarding trunk measurements and the muscle test, applied according to the MRC Scale.

## 4. Discussion

Global Postural Re-education (RPG) can be crucial for Parkinson’s patients, as it addresses postural issues like camptocormia, positively impacting balance, mobility, ADLs, and QoL. For this reason, the aim of this study was to verify if postural gymnastics, according to the Souchard method, are effective in terms of camptocormia, posture, trunk mobility, balance, QoL, and ADLs.

According to the statistical data, significant results were obtained. As reported in Table 1, PDQ-39 mobility and total score improvements were observed and maintained over time; ADLs and emotional well-being improved at the end of treatment and after one month (t1) but can no longer be seen at 3 months (t3). On the other hand, stigma and bodily discomfort score significantly improved three months after the treatment. Similar results were obtained by Paolucci et al. [21] in a study where the Mezières method was applied: the treatment group showed significant improvements in terms of QoL based on two subscales of the Short Form Health Survey 36 (SF-36) after 12 weeks, in contrast to the control group [42].

Balance was assessed by administering the Berg Balance scale (Table 2); improvements were shown at the end of treatment (t1) and maintained even at three months (t3). This result is consistent with another study comparing the efficacy of kinesio taping versus GPR [43]: in patients treated with GPR, greater improvement in BBS were shown 1 month after treatment, but the same scores were not maintained at the second follow-up (2 months post treatment). As reported in the previously mentioned studies, GPR aims to improve awareness of one’s body and the movement associated with it, tasks largely performed by the proprioceptive system. The feedback received from muscles, tendons, and joints during postural exercises helps to acquire awareness of postural habits, providing concrete strategies to actively correct them and helping to obtain better motor control and balance [22].

Scientific studies show that a lack of flexibility in the spine contributes to the development of balance and motor control deficits in patients with PD [44]; at the same time, it reduces the performance of many postural (e.g., rolling over, changes in direction during walking) and daily life activities [45]. In this regard, particular attention was paid to measuring trunk range of motion before and after treatment, and statistically significant results have been obtained, as reported in Table 3. Improvements were shown in flexion, extension, right-side bending, left-side bending, right rotation and left rotation from the end of the treatment (t1) to the second follow up (t3), while the improvement of the neutral position from C7 to L5 is lost after the one and three month follow ups. The obtained results coincide with the studies of Bartolo et al. (2010) [46] and Stożek et al. (2016) [44] in which intensive rehabilitation programs (e.g., postural re-education, balance exercises, gait training, etc.) have achieved positive effects even in the long term (4 weeks). Moreover, Table 4 reports a statistically significant improvement in the strength of the rectus abdominis and the back extensor, according to the MRC scale, from the end of rehabilitation (t1) to the 3 months follow-up (t3). Muscles like the rectus abdominis play a crucial role in the correction of posture and balance [40,41]; improvements in its functionality can help in facilitating core stability, stabilizing the trunk, and promoting the acquisition of balance and movement control. On the contrary, the back extensor muscles are responsible for maintaining the upright position of the spine, influencing trunk control and the maintenance of an upright posture. These two muscle groups work synergistically to improve postural alignment and promote functional independence in people with Parkinson’s Disease, counteracting postural rigidity and positively influencing mobility and QoL.

It is essential to consider the long-term management of patients with camptocormia with continuous monitoring and adjustments in the proposed treatment, as this is necessary to maintain the achieved benefits over time against the progression of the disease.

## 5. Limitations of the Study

This study presents limitations that could impact the results’ interpretation and generalizability. Firstly, the patient sample was small and may not represent the entire Parkinson’s patient population; additionally, expanding the number of selected patients is advisable to obtain more reliable results and enable a more significant assessment of outcomes. Another limitation is the follow-up periods, with a short observation of the effects of GRP. These limitations should be taken into consideration when evaluating the applicability of the results and may indicate the need for further research to comprehensively address the validity of treatment for camptocormia in Parkinson’s patients. Further studies are needed to recruit more patients and carry out longer follow-up period checks to verify the actual effectiveness of the proposed intervention.

## 6. Conclusions

In conclusion, from the following study, postural gymnastic treatment according to Souchard may offer promising benefits for camptocormia-afflicted PD patients. The method has shown the potential to improve mobility, static and dynamic balance and trunk flexibility, as well as QoL and ADLs. However, some challenges and issues must be addressed. Long-term management of patients with camptocormia is needed and requires an individualized approach, with a continuous focus on assessment and treatment adaptation based on each patient’s specific needs. These results underline the need for further research and development in this field to optimize treatment effectiveness and expand therapeutic options.

These preliminary results highlight the importance of continued research on postural interventions to optimize existing strategies. Furthermore, considering the wide clinical variability of patients with PD, it must be emphasized that postural gymnastics should not be administered alone but rather included within the numerous rehabilitation therapies necessary for this condition.

## Figures and Tables

**Figure 1 jcm-13-06166-f001:**
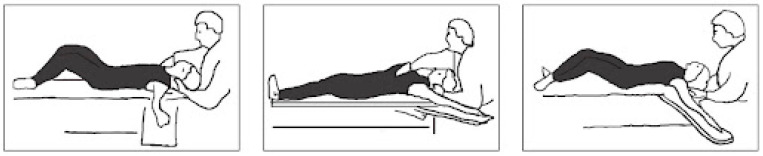
The frog on the ground posture and progression.

**Figure 2 jcm-13-06166-f002:**
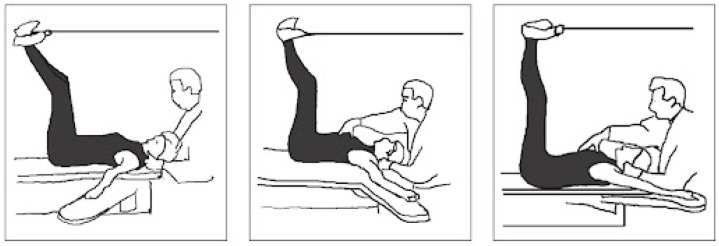
The frog on the air posture and progression.

**Table 1 jcm-13-06166-t001:** Data for the PDQ-39 scale.

PDQ-39	T0		T1		Z	*p*	T2		Z	*p*	T3		Z	*p*
	Mean ± SD	Median	Mean ± SD	Median			Mean ± SD	Median			Mean ± SD	Median		
Mobility	6.78 ± 7.2	4	4.22 ± 5.1	2	−2.201	**0.028 ***	3.33 ± 4	2	−2.207	**0.027 ***	2.33 ± 2.5	2	−2.375	**0.018 ***
ADL	4.11 ± 4.5	2	2.11 ± 2.8	1	−2.232	**0.026 ***	1.56 ± 2	1	−2.207	**0.027 ***	2.22 ± 3.0	0	−1.382	0.167
Emotional well-being	4.44 ± 3.8	4	2.56 ± 2.1	2	−1.973	**0.049 ***	2.00 ± 1.5	2	−2.108	0.035 *	2.11 ± 1.5	2	−1.62	0.105
Stigma	1.67 ± 2.5	1	0.56 ± 1.3	0	−1.89	0.059	0.22 ± 0.4	0	−1.89	0.059	0	0	−2.06	**0.039 ***
Social Support	0.33 ± 1.0	0	0.56 ± 1.1	0	−1	0.317	0.00 ± 0.0	0	−1	0.317	0.22 ± 0.6	0	−0.447	0.655
Cognition	4.11 ± 3.1	3	3.56 ± 2.5	3	−1.095	0.273	3.00 ± 2.3	3	−1.801	0.072	2.67 ± 2.3	3	−1.851	0.064
Communication	1.33 ± 1.9	0	0	0	−1.826	0.068	0.00 ± 0.0	0	−1.826	0.068	0.00 ± 0.0	0	−1.826	0.068
Bodily discomfort	3.22 ± 2.7	4	2.22 ± 1.9	2	−1.594	0.111	2.22 ± 1.3	3	−1.538	0.124	1.78 ± 1.9	2	−2.214	**0.027 ***
Tot	26.00 ± 18	23	15.78 ± 12.4	11	−2.666	**0.008 ***	12.33 ± 8.1	11	−2.666	**0.008 ***	11.33 ± 8	14	−2.429	**0.015 ***

SD: Standard Deviation. * *p* < 0.05.

**Table 2 jcm-13-06166-t002:** Data for the BBS.

BBS	Mean ± SD	Median	Mean ± SD	Median	Z	*p*
T0-T1	47.33 ± 6.614	52.00	53.22 ± 3.346	55.00	−2.677	**0.007 ***
T0-T2	47.33 ± 6.614	52.00	54.00 ± 2.398	55.00	−2.670	**0.008 ***
T0-T3	47.33 ± 6.614	52.00	54.33 ± 2.500	56.00	−2.675	**0.007 ***

SD: Standard Deviation. * *p* < 0.05.

**Table 3 jcm-13-06166-t003:** Data for trunk measurement.

Trunk	T0		T1		Z	*p*	T2		Z	*p*	T3		Z	*p*
Measurement	Mean ± SD	Median	Mean ± SD	Median			Mean ± SD	Median			Mean ± SD	Median		
Neutral from c7 to l5	49.33 ± 4.3	50	48.11 ± 4.4	49	−2.156	**0.031 ***	48.33 ± 4.3	50	−1.913	0.056	48.44 ± 4.5	50	−1.947	0.052
Flexion	56.00 ± 4.2	57	58.00 ± 3.8	58	−2.701	**0.007 ***	58.22 ± 3.9	58	−2.687	**0.007 ***	58.11 ± 3.8	58	−2.539	**0.011 ***
Extension	46.22 ± 5.1	46	43.78 ± 4.9	45	−2.694	**0.007 ***	44.00 ± 4.8	45	−2.555	**0.011 ***	44.11 ± 4.9	45	−2.379	**0.017 ***
Neutral side bending right	62.33 ± 4.7	61	62.67 ± 3.9	62	−0.718	0.473	62.56 ± 4.0	62	−0.568	0.57	62.89 ± 4.3	62	−1.065	0.287
Neutral side bending left	62.11 ± 4.7	61	62.33 ± 4.3	62	−0.493	0.622	62.78 ± 4.2	62	−0.933	0.351	62.78 ± 4.4	62	−1.436	0.151
Right-side Bending	53.11 ± 5.6	55	48.56 ± 5.2	49	−2.677	**0.007 ***	47.78 ± 4.6	47	−2.692	**0.007 ***	48.00 ± 4.5	49	−2.684	**0.007 ***
Left-side bending	52.11 ± 5.6	50	47.56 ± 5.2	49	−2.67	**0.008 ***	47.00 ± 4.5	47	−2.527	**0.012 ***	47.22 ± 4.6	48	−2.527	**0.012 ***
Neutral right rotation	77.89 ± 6.4	80	76.56 ± 5.6	78	−1.594	0.111	77.00 ± 6.0	80	−0.843	0.399	77.00 ± 6.0	80	−0.843	0.399
Neutral left rotation	78.22 ± 6.0	80	77.11 ± 5.9	78	−1.633	0.102	77.89 ± 6.3	80	−0.513	0.608	77.89 ± 6.2	80	−0.531	0.595
Right rotation	81.00 ± 6.2	84	84.11 ± 5.9	87	−2.692	**0.007 ***	84.78 ± 5.9	87	−2.677	**0.007 ***	84.67 ± 5.9	87	−2.68	**0.007 ***
Left rotation	81.78 ± 5.8	84	84.78 ± 5.9	87	−2.68	**0.007 ***	85.11 ± 5.9	87	−2.694	**0.007 ***	85.11 ± 6.4	88	−2.689	**0.007 ***

SD: Standard Deviation. * *p* < 0.05.

**Table 4 jcm-13-06166-t004:** Data for muscle test according to the MRC scale.

	T0		T1		Z	*p*
	Mean ± SD	Median	Mean ± SD	Median		
Rectus Abdominis	4.33 ± 0.707	4.00	4.78 ± 0.441	5.00	−2.000	**0.046 ***
Internal Obliques	3.44 ± 0.527	3.00	3.56 ± 0.527	4.00	−1.000	0.317
External Obliques	3.44 ± 0.527	3.00	3.56 ± 0.527	4.00	−1.000	0.317
Back Extensor	3.67 ± 1.225	4.00	4.22 ± 1.093	5.00	−2.236	**0.025 ***
	**T0**		**T1**		**Z**	** *p* **
Rectus Abdominis	4.33 ± 0.707	4.00	4.78 ± 0.441	5.00	−2.000	**0.046 ***
Internal Obliques	3.44 ± 0.527	3.00	3.56 ± 0.527	4.00	−1.000	0.317
External Obliques	3.44 ± 0.527	3.00	3.56 ± 0.527	4.00	−1.000	0.317
Back Extensor	3.67 ± 1.225	4.00	4.22 ± 1.093	5.00	−2.236	**0.025 ***
	**T0**		**T1**		**Z**	** *p* **
Rectus Abdominis	4.33 ± 0.707	4.00	4.78 ± 0.441	5.00	−2.000	**0.046 ***
Internal Obliques	3.44 ± 0.527	3.00	3.67 ± 0.707	4.00	−1.414	0.157
External Obliques	3.44 ± 0.527	3.00	3.67 ± 0.707	4.00	−1.414	0.157
Back Extensor	3.67 ± 1.225	4.00	4.22 ± 1.093	5.00	−2.236	**0.025 ***

SD: Standard Deviation. * *p* < 0.05.

## Data Availability

The data that support the findings of this study are available from the corresponding author upon reasonable request.

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
