# Peer review of "An Exploratory Study on the Effects of Souchard Postural Gymnastics in Parkinson’s Disease Patients with Camptocormia: A Quasi-Experimental Approach"

_jcm, 2024, doi:10.3390/jcm13206166_

Round 1
Reviewer 1 Report
Comments and Suggestions for Authors
Firstly, I would like to thank the authors for submitting their interesting and relevant research. I have a few suggestions bellow:
1) It is advisable to decrease the word count in the title. The study design should be present in the title as the authors wrote.
2) We suggest working with English editing services to improve the readability of your article
3) In the abstract, I recommend rewriting L14 where authors state that camptocormia is a sign of PD. Camptocormia is a syndrome and it could be associated with PD, but is also associated with several other conditions and drugs as well.
4) L17: please clarify “management of the trunk”. Do authors mean truncal weakness?
5) Please state the study hypothesis in the abstract and in the last paragraph of the introduction
6) Please include epidemiological data about PD in percentages
7) Please include clinical diagnostic criteria for PD, we recommend including Neurology journal latest diagnostic criteria
8) We recommend having your article reviewed by a movement disorders specialist. There are some definitions that’s should be revised, Pisa syndrome is different of camptocormia, PD definitions should be revised as well
9) Please explain if scales used were validated for your study
10) Please explain why this particular design was chosen
11) Please further explain inclusion and exclusion criteria for this study
12) Please explain how was the n of the study calculated
13) Please explain why a randomized controlled trial was not done
14) Please review your methods and statistical analysis section with a professional statistician and properly report analysis performed
15) Improve your results and discussion to reflect your study design
16) Please provide IRB approval
17) Please restructure your article according with TREND reporting guidelines and offer a point by point explanation of how the guidelines were followed
Haynes AB, Haukoos JS, Dimick JB. TREND reporting guidelines for nonrandomized/quasi-experimental study designs. JAMA surgery. 2021 Sep 1;156(9):879-80.
Comments on the Quality of English Language
Extensive editing required.
Author Response
See the document attached below.

Reviewer 2 Report
Comments and Suggestions for Authors
Here are my critiques:
- Sample size: The study has a very small sample size of only 9 participants. This limits the statistical power and generalizability of the results. The authors acknowledge this as a limitation, but it's a significant weakness.
- Lack of control group: There is no control or comparison group, making it difficult to attribute improvements solely to the intervention rather than other factors.
- Short follow-up period: The longest follow-up is only 3 months, which is relatively short for assessing long-term effects in a chronic progressive disease like Parkinson's.
- Limited scope: The study focuses only on patients with early-stage Parkinson's (Hoehn and Yahr scale ≤2). The effectiveness for more advanced cases is not addressed.
- Potential bias: There's no mention of blinding of assessors or participants, which could introduce bias.
- Statistical approach: The study uses multiple comparisons without apparent correction for familywise error rate, which increases the risk of Type I errors.
- Inconsistent results: Some improvements (e.g., in ADLs and emotional well-being) were not maintained at 3 months, raising questions about the long-term efficacy of the intervention.
- Limited explanation of mechanism: While the study shows some improvements, there's limited discussion on the physiological mechanisms by which the intervention might work.
- Lack of adverse event reporting: The manuscript doesn't mention whether any adverse events occurred during the study.
- Incomplete literature review: The introduction could benefit from a more comprehensive review of existing literature on postural interventions in Parkinson's disease.
- Unclear randomization: It's not specified whether there was any randomization in patient selection or treatment allocation.
- Limited generalizability: The study was conducted at a single center, which may limit its generalizability to other settings or populations.
- Incomplete reporting of methods: Some details of the intervention (e.g., exact duration of postures) are not provided.
- Lack of cost-effectiveness analysis: Given the intensive nature of the intervention (10 biweekly sessions), some consideration of cost-effectiveness would be valuable.
- Overstatement of conclusions: Given the limitations, the conclusions may be overstated. More cautious language acknowledging the preliminary nature of the findings would be appropriate.
Author Response
See the file attached below.

Round 2
Reviewer 1 Report
Comments and Suggestions for Authors
The authors have improved the manuscript and it is now ready to be published
Author Response
Thank you for your approval!
Reviewer 2 Report
Comments and Suggestions for Authors
The authors responded to my comments well. However, due to the the small sample size and the resulting interpretation of the results, I suggest to change the title to "An Exploratory Study on the Effects of Souchard Postural Gymnastics in Parkinson's Disease Patients with Camptocormia: A Quasi-Experimental Approach".
Author Response
Thanks for your constructive comments. Changes have been made to the article title as suggested.